# Anosognosia for hemiplegia as a tripartite disconnection syndrome

Valentina Pacella[1,2]*, Chris Foulon[3,4,5], Paul M Jenkinson[6], Michele Scandola[2], Sara Bertagnoli[2], Renato Avesani[7], Aikaterini Fotopoulou[8†], Valentina Moro[2†], Michel Thiebaut de Schotten[3,4,9†]*

[1]Social and Cognitive Neuroscience Laboratory, Department of Psychology, Sapienza University of Rome, Rome, Italy; [2]NPSY.Lab-VR, Department of Human Sciences, University of Verona, Verona, Italy; [3]Brain Connectivity and Behaviour Laboratory, Sorbonne Universities, Paris, France; [4]Frontlab, Institut du Cerveau et de la Moelle épinière (ICM), UPMC UMRS 1127, Inserm U 1127, CNRS UMR 7225, Paris, France; [5]Computational Neuroimaging Laboratory, Department of Diagnostic Medicine, The University of Texas at Austin Dell Medical School, Austin, United States; [6]School of Life and Medical Sciences, University of Hertfordshire, Hatfield, United Kingdom; [7]Department of Rehabilitation, IRCSS Sacro Cuore-Don Calabria Hospital, Verona, Italy; [8]Clinical, Educational and Health Psychology, Division of Psychology and Language Sciences, University College London, London, United Kingdom; [9]Groupe d'Imagerie Neurofonctionnelle, Institut des Maladies Neurodégénératives-UMR 5293, CNRS, CEA University of Bordeaux, Bordeaux, France

*For correspondence:
valentina.pacella.90@gmail.com
(VP);
michel.thiebaut@gmail.com (MTS)

†These authors contributed equally to this work

Competing interests: The authors declare that no competing interests exist.

**Abstract** The syndrome of Anosognosia for Hemiplegia (AHP) can provide unique insights into the neurocognitive processes of motor awareness. Yet, prior studies have only explored predominately discreet lesions. Using advanced structural neuroimaging methods in 174 patients with a right-hemisphere stroke, we were able to identify three neural systems that contribute to AHP, when disconnected or directly damaged: the (i) premotor loop (ii) limbic system, and (iii) ventral attentional network. Our results suggest that human motor awareness is contingent on the joint contribution of these three systems.
DOI: https://doi.org/10.7554/eLife.46075.001

## Introduction

Motor awareness allows individuals to have insight into their motor performance, a fundamental aspect of self-awareness. However, following brain damage, some patients may fail to acknowledge their contralesional paralysis, even after this has been repeatedly demonstrated to them. This refractory (delusional) unawareness of motor impairments is termed anosognosia for hemiplegia (AHP, *Babinski, 1914*). The syndrome is usually reported in right hemisphere lesions, although in more recent years the possibility of motor awareness deficits following left hemisphere lesions has been advanced (*Cocchini et al., 2009*). The syndrome is reported to be relatively frequent after right hemisphere damage in the very acute phase after lesion onset (32% rate) but this usually resolves in the first weeks (18% rate within the first week and 5% rate at 6 months; *Vocat et al., 2010*). Studying AHP offers unique opportunities to explore the neurocognitive mechanisms of motor awareness.

Early studies regarded AHP as secondary to other concomitant symptoms (*Cocchini et al., 2009*; *Vocat et al., 2010*; *Levine, 1990*; *Karnath et al., 2005*), in particular spatial deficits such as hemineglect (*Bisiach, 1999*) caused by parietal lesions. More recent experimental and voxel-based, lesion-

symptom mapping (VLSM) results suggest that AHP is an independent syndrome. These studies address AHP as an impairment of action and body monitoring, with lesions to the lateral premotor cortex (*Berti et al., 2005*) and the anterior insula (*Karnath et al., 2005*), affecting patients' ability to detect discrepancies between feed-forward motor predictions and sensorimotor feedback. However, these hypotheses are insufficient to explain all the AHP symptoms, such as patients' inability to update their delusional beliefs based on social feedback or more general difficulties experienced in their daily living (*Fotopoulou, 2014*; *Vuilleumier, 2004*). Indeed, others have suggested that AHP can be caused by a functional disconnection between regions processing top-down beliefs about the self and those processing bottom-up errors regarding the current state of the body (*Fotopoulou, 2014*; *Mograbi and Morris, 2013*). Nevertheless, to date the brain disconnection hypothesis could not be explored due to the relatively small sample size and the standard methodology of previous studies, which favours the implication of discreet lesion locations in the pathogenesis of AHP.

Here, to overcome this gap, we took advantage of (i) the largest cohort of AHP patients to date (N = 174; 95 hemiplegic and AHP patients diagnosed by *Bisiach et al., 1986* and 79 hemiplegic controls) and (ii) an advanced lesion analysis method (BCBtoolkit, *Foulon et al., 2018*). This method generates a probabilistic map of disconnections from each patient's brain lesion to identify the disconnections that are associated with given neuropsychological deficits at the group level. Previous use of this connectivity approach has already proven fruitful in the study of neuropsychological deficits (*Thiebaut de Schotten et al., 2014*; *Thiebaut de Schotten et al., 2015*; *Fox, 2018*).

We predicted that AHP would be associated not only with focal grey matter lesions, but also with long-range disconnections due to the white matter damage, in particular to tracts associated with sensorimotor monitoring and self-reflection. Specifically, we anticipated the possibility that motor awareness emerges from the integrated activation of separated networks (*Luria, 1966*; *Shine et al., 2019*), whose contributions feed into the multifaceted expression of the syndrome.

## Results

To test these predictions, we first conducted anatomical investigations to identify lesion sites and created probability maps of white matter tracts' disconnection. These results were statistically analysed by means of two regression analyses, to identify the contribution of grey and white matter structures in AHP, considering differences in age, lesion size, lesion onset-assessment interval and critical motor and neuropsychological deficits (i.e. covariates of control). Considering our sample size and a power of 95%, t values above two correspond to a medium effect size (cohen d > 0.5) and t values above 3.6 correspond to a large effect size (cohen d > 0.8).

The regression computed on the lesion sites (*Figure 1a*) indicated the involvement of grey matter structures previously associated with AHP (*Moro et al., 2016*), such as the insula (anterior long gyrus, t = 4.89; p=0.002), the temporal pole (t = 4.77; p=0.003), and the striatum (t = 4.68; p=0.003) as well as a very large involvement of white matter (t = 4.98; p=0.002). The second regression, on white matter maps of disconnection (*Figure 1b*), revealed a significant contribution of the cingulum (t = 3.85; p=0.008), the third branch of the superior longitudinal fasciculus (SLF III; t = 4.30; p=0.003), and connections to the pre-supplementary motor area (preSMA; t = 3.37; p=0.013), such as the frontal aslant and the fronto-striatal connections. No other tracts or structures were significantly involved in AHP.

To test whether AHP emerges from the damage to grey matter structures and disconnection of each of these white matter tracts independently or together as a whole, we first investigated their contribution pattern to AHP by means of Bayesian computation of generalised linear multilevel models. 100 binomial models were computed to take into account the potential contribution of clinical/demographic effects, disconnected tracts and lesioned grey matter structures to AHP, starting from the clinical/demographics model (with only the control covariates, that is age, education, lesion size, lesion onset-assessment interval and critical motor and neuropsychological deficits) to the full model, with all the control covariates, the grey matter structures, the tracts, and all the interactions among them (*Gelman and Hill, 2006*; see Materials and methods section). The results indicated positive support for the striatum ($BF_{10}$ = 3.22) and weak support for the insula ($BF_{10}$ = 1.22) and the temporal pole ($BF_{10}$ = 2.23) to AHP. Together, these three grey matter structures showed strong support to AHP ($BF_{10}$ = 150). In the white matter, the disconnection of each tract was critical to AHP (Cingulum, $BF_{10}$ = 270.98; FST, $BF_{10}$ = 180.48; FAT, $BF_{10}$ = 367.61; SLF III, $BF_{10}$ = 571.49). No other tracts

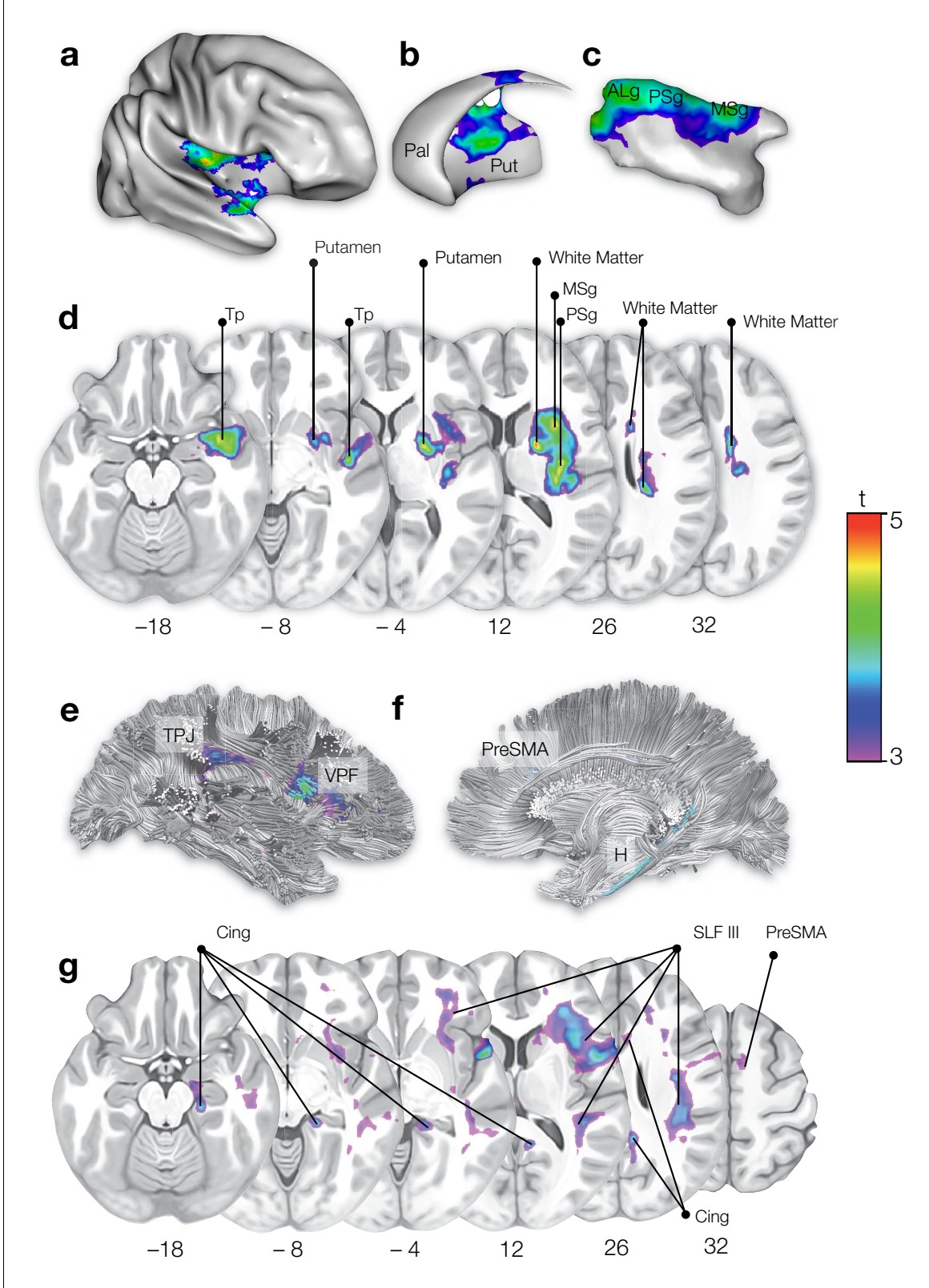

**Figure 1.** On the top half, statistical mapping of the lesioned areas in AHP. (a) right hemisphere (b) striatum (c) insula (d) axial sections. Pal: pallidum; Put: putamen; ALg: anterior long gyrus; PSg: posterior short gyrus; MSg: middle short gyrus; Tp: temporal pole. On the bottom half, statistical mapping of the brain disconnections in AHP. (e) right hemisphere lateral view; (f) right hemisphere medial view; (g) axial sections. TPJ: temporo-parietal junction;

*Figure 1 continued on next page*

*Figure 1 continued*

VPF: ventral prefrontal cortex; preSMA: pre-supplementary area; H: hippocampus; Cing: cingulum; SLF III: third (ventral) branch of the superior longitudinal fasciculus; PreSMA: pre-supplementary motor area.

DOI: https://doi.org/10.7554/eLife.46075.002

contributed significantly to AHP. Additionally, results indicated that the model that included the contribution of all the four tracts better predicts AHP when compared to direct grey matter lesions ($BF_{12} > 150$). However, the model that best fits with our data included the contribution of all the four tracts and grey matter structures ($BF_{10} > 150$) supporting the hypothesis that the joint contribution of both disconnections and direct lesions is the best predictor of AHP (*Figure 2*).

It is worth noting that although the starting model showed that the clinical and demographic variables alone contribute to the AHP symptoms, the model with only these control variables was less efficacious in predicting AHP: i) when compared to the models that included each of the tracts resulting from the disconnection analysis (Cingulum, SLF III, FAT and FST); ii) when compared to the model that included the interaction among the four tracts and iii) when compared to the model that included also the grey matter structures resulting from the lesion analysis (i.e. insula, Putamen and temporal pole, the white +grey matter model; BF >150). We can thus consider that, although the severity of concomitant cognitive deficits may contribute to AHP (*Cocchini et al., 2009*; *Vocat et al., 2010*; *Levine, 1990*; *Karnath et al., 2005*), the syndrome is fully explained only when considering the integrated contribution of specific neural networks.

These results, derived from the largest lesion mapping study on AHP to date, show that lesions and white matter disconnections in three networks contribute to AHP: (i) posterior parts of the limbic network (i.e. cingulum connections between the amygdala, the hippocampus and the cingulate gyrus); (ii) the ventral attentional network (i.e. SLF III connections between temporo-parietal junction and ventral frontal cortex); and (iii) the premotor loop (i.e. frontal aslant and fronto-striatal connections between the striatum, the preSMA and the inferior frontal gyrus).

## Discussion

Previous lesion mapping studies in AHP have highlighted the role of discrete cortical lesions in areas such as the lateral premotor cortex or the insula, and suggested corresponding theories of motor and body awareness (*Karnath et al., 2005*; *Berti et al., 2005* and *Fotopoulou, 2014*; *Vuilleumier, 2004* for critical reviews). By contrast, our results suggest a major role of white matter disconnection in AHP that, when combined to the direct grey matter lesions, reveals that AHP is a tripartite disconnection syndrome involving disruptions in tracts and structures belonging to three systems: the pre-motor loop, the limbic system and the ventral attentional network. Correspondingly, motor awareness requires the convergence (i.e. integration) of a number of cognitive processes, rather than being a purely motor monitoring function. Indeed, this interpretation is consistent with the delusional features of the syndrome (see *Fotopoulou et al., 2010*) and many experimental findings in AHP, showing these patients have difficulties to update beliefs (*Vocat et al., 2013*), take allocentric perspectives (*Besharati et al., 2016*) and conduct reality monitoring (*Jenkinson et al., 2010*) beyond the motor domain.

The observed direct damage to and disconnections of the pre-motor network (pre-SMA, striatum and inferior frontal gyrus) support the hypothesis of difficulties in monitoring motor signals and learning from action failures ('did you execute the action? Yes, I did'; *Fotopoulou et al., 2008*; *Berti et al., 2005*). However, more general processes associated with the sense of self are damaged in AHP, such as the top-down beliefs about the self and the processing of bottom-up errors regarding the current state of the body. Specifically, the well-documented deficits in AHP patients' general awareness ('why are you in hospital?'), anticipatory awareness ('are you able to reach the table with your left hand?'; *Moro et al., 2015*) and mentalisation ('the doctors think there is some paralysis, do you agree?'; *Besharati et al., 2016*; *Feinberg, 2000*) might be explained by disconnections of the limbic system structures via the cingulum. In fact, the limbic system has been previously associated with emotional and memory processing and is part of the default mode network (*Greicius et al., 2009*) a pattern of intrinsic connectivity observed during self-referential, introspective states, including autobiographical retrieval, future imaging and mentalisation.

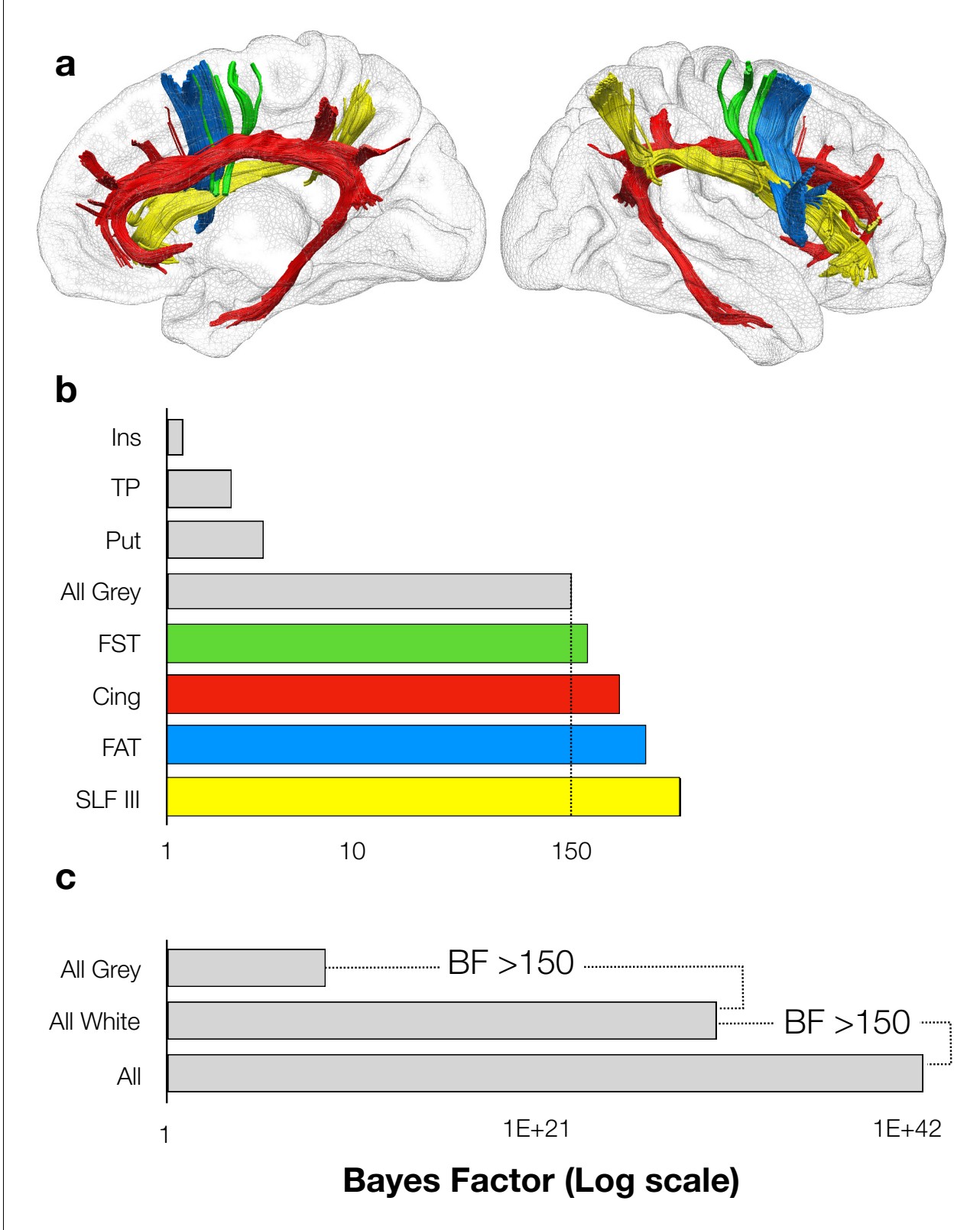

**Figure 2.** Motor awareness network. (**a**) right hemisphere medial view (left) right hemisphere lateral view (right); (**b-c**) Bayes Factors for all models, each one representing the hypothesis that the damage to grey matter structure and/or the tract disconnection is necessary to explain AHP, against the clinical/demographic model. Ins: insula; TP: temporal pole; Put: putamen; FST: fronto-striatal tract; Cing: cingulum; FAT: frontal aslant tract; SLF III: third branch of the superior longitudinal fasciculus.

*Figure 2 continued on next page*

*Figure 2 continued*

DOI: https://doi.org/10.7554/eLife.46075.003

Finally, the disconnection of the ventral attentional network (i.e. via the SLF III connections in the right hemisphere), would prevent the possibility to appreciate the salience of stimuli referring to one's own paralysis (*Corbetta et al., 2008*), as also suggested by previous experimental manipulations of emotions and arousal in AHP (*Besharati et al., 2014*; *D'Imperio et al., 2017*). It is worth mentioning that the insula (that has been previously found critical in AHP) contributes to both the limbic system and the ventral attentional network and has an important role in the updating of self-referred beliefs by integrating external sensory information with internal emotional and bodily state signals (*Craig, 2009*). In sum, damage to the integrated contribution of the aforementioned three networks makes it difficult to update beliefs regarding the bodily self in on-line, motor tasks ('I did not move my leg now'), as well as more generally ('I cannot walk as I have always done').

Importantly, no isolated pattern of lesions or disconnections to any of these systems (i.e. the limbic system, the ventral attentional network, or the premotor system), or any demographical and clinical variables can fully explain AHP (*Figure 2*). We thus postulate that deficits in motor monitoring, associated with a compromised premotor network, need to be combined with other salience and belief updating deficits, collectively leading to a multifaceted syndrome in which premorbid beliefs and emotions about the non-paralysed self dominate current cognition about the paralysed body. These results open up interesting hypotheses on the hierarchical or parallel relations between the three networks, in terms of temporal activations (either serial, parallel or recurrent) that remain to be explored in future studies.

The main limitations of the study are related to manual lesion delineation and registration methods (*de Haan and Karnath, 2018*; *Rorden and Karnath, 2004*) and the sensitivity level of neuroimaging techniques that do not depict the full extent of damage produced by stroke lesions (*Hillis et al., 2000*). However, these limitations mainly apply to small sample studies, while here the large number of patients investigated reduces these risks. Furthermore, only right hemisphere damaged patients were analysed in the study, due to the classical association of the syndrome with right lateralized lesions. Further studies are needed to investigate the neuroanatomical correlates of anosognosic symptoms recently reported following left hemisphere lesions.

In conclusion, on the basis of a large (N = 174) and advanced grey and white matter lesion-mapping study, we demonstrate a tripartite contribution of lesions and disconnections to the pre-motor network, the limbic system, and the ventral attentional network to motor unawareness. We thus suggest that motor awareness is not limited to sensorimotor monitoring but also requires the joint contribution of cognitive processes involved in maintaining and updating beliefs about self.

## Materials and methods

### Design and statistical analysis

The aim of the study was to explore the white matter disconnections involved in AHP. To this end, we investigated the neural systems that contribute to the symptoms of AHP. To the best of our knowledge, this approach has never been applied to the study of AHP and it can shed light on the theoretical and phenomenological complexity of the disease, by integrating and going beyond existing findings gained through classic lesion studies (*Vocat et al., 2010*; *Karnath et al., 2005*; *Berti et al., 2005*; *Moro et al., 2016*; *Fotopoulou et al., 2010*; *Moro et al., 2011*).

For this purpose, we collected neuroimaging and clinical data from a large sample of right hemisphere stroke patients. To compute lesion sites and map of disconnections that were strictly related to the AHP pathology, we compared our target group of AHP patients with a group of stroke patients with hemiplegia but without AHP. Patients' map of lesions and disconnections were statistically compared between the two groups and adjusted for control covariates. In fact, variance related to patients' demographic variables (age and education level) was removed. As previous lesion studies (*Vocat et al., 2010*; *Moro et al., 2016*) found some differences in neuronal correlates of AHP in acute and chronic stages, the interval between lesion onset and neuropsychological assessment was

controlled as well. We also included the lesions' size of our sample, taking into account the number of voxels of each lesion as a nuisance variable.

Finally, when computing the AHP map of lesions and disconnections we controlled for the clinical (onset-assessment interval, motor deficits) and neuropsychological (personal, extra-personal neglect and memory impairments) symptoms that are often associated with AHP but are related to different patterns of disconnection.

The tracts emerging from this analysis were further analysed by means of Bayesian models to confirm the individual involvement of each tract and test their joint contribution to AHP (see details below).

## Patients

Data from 195 stroke patients with unilateral right hemisphere damage were collected from two collaborating centers based in Italy and the United Kingdom over a period of 10 years.

Patients' inclusion criteria were: (i) unilateral right hemisphere damage, secondary to a first-ever stroke, as confirmed by clinical neuroimaging; (ii) severe plegia of their contralateral upper limb (AHP left arm, MRC $\leq$ 2), as clinically assessed (MRC scale; *Matthews, 1976*). Exclusion criteria were: (i) previous history of neurological or psychiatric illness; (ii) medication with severe cognitive or mood side-effects; (iii) severe language, general cognitive impairment, or mood disturbance that precluded completion of the study assessments.

The MRI or CT neuroimaging data were available for 174 out of 195 patients. They were divided into two groups according to the presence/absence of AHP (see below for AHP assessment details), resulting in a group of 95 AHP patients and 79 non-AHP, hemiplegic control (HP) subjects. Among these, clinical and anatomical data of 40 AHP patients and 27 controls has been described in a previous study (*Moro et al., 2016*). Groups were balanced for demographic data (age, education, interval period between lesion onset and assessments) and lesion size. As the data were collected from different stroke recovery units, we took into account the neurological and neuropsychological tests that were most commonly administered to all the patients across the different centers (*Table 1*).

All patients gave written, informed consent and the research was conducted in accordance with the guidelines of the Declaration of Helsinki (2013) and approved by the Local Ethical Committees

**Table 1.** For AHP and control groups, mean and (±standard deviation) of demographic and clinical variables, neurological and neuropsychological assessments are reported.

| | Ahp (N = 95) | Hp (N = 79) |
|---|---|---|
| **Demographic and clinical** | | |
| Age (years) | 68.48 ± 12.54 | 63,01 ± 13.49 |
| Education (years) | 9.46 ± 3.74 | 11 ± 3.77 |
| Interval (days) | 35.74 ± 40.58 | 44.42 ± 46.7 |
| Lesion Size (voxels) | 134327.74 ± 113196.17 | 113082.73 ± 120844.22 |
| **Anosognosia** | | |
| Bisiach score | 2.46 ± 0.6 | 0 ± 0 |
| **Personal neglect** | | |
| Comb$\left(\frac{left-right\ strokes}{left+ambiguous+right\ strokes}\right)$ | −0.3 ± 0.4 | −0.06 ± 0.47 |
| **Extra-personal neglect** | | |
| Line cancellation (number of items cancelled) | 19.26 ± 11.9 | 28.35 ± 10.77 |
| **Memory Span** | | |
| Digit/verbal span (number of items recalled) | 5.65 ± 2.14 | 6.83 ± 2.46 |
| **Motor index** | | |
| MRC (LUL) | 0.15 ± 0.42 | 0.6 ± 0.99 |

DOI: https://doi.org/10.7554/eLife.46075.004

of each center (Comitato Etico per la sperimentazione clinica delle province di Verona e Rovigo, Proj. 602CESC, Prot. 47566 frl 14/10/2015; National Health System Research Ethics Committee, with ref no: 05/Q0706/218 and 04/Q2602/77).

## Neurological and neuropsychological assessment

Patients were identified as anosognosic or control according to their score in the Bisiach scale (*Bisiach et al., 1986*). This investigates the explicit form of awareness related to one's limb paralysis. During the scale administration, patients were required to verbally answer a 4-point interview about their current condition: a '0' score indicates a spared consciousness of the disease (=the disorder is spontaneously reported or mentioned by the patient following a general question about his/her complains), a '1' score is assigned when patients refer to their disability only after specific questions about the strength of their left limbs, while patients scoring '2' or '3' are considered anosognosic for their awareness of the disease emerging only after a demonstration through a routine technique of neurological examination (score 2) or not emerging at all (score 3). In the study, all the patients in the AHP group had a score $\geq$2, while those in the HP group had a score = 0.

Plegia of the contralesional upper limb was assessed through the Medical Research Council 5-point scale (*Matthews, 1976*), ranging from 5 (normal functioning) to 0 (no movement). Only patients with a score $\leq$2 were included in the study.

Personal neglect was assessed by means of the 'Comb' test, from the 'Comb/Razor test' (*McIntosh et al., 2000*). We referred to each patient's score on the line cancellation subtest of the BIT as our measure of extra-personal neglect (Behavioural Inattention Test, *Wilson et al., 1987*). Finally, we used the digit/word span (*Baddeley et al., 1975*; *Baddeley, 1996*) to assess working memory. The 3-nearest neighbour computation replaced the missing data from the demographic and clinical variables (education: 4.9%; lesion-assessment interval: 0.05%; motricity index: 1.9%; personal neglect: 1.7%; extra-personal neglect: 2.5%; memory span: 6.2%).

In order to compare results expressed in different scoring ranges, all the scores from neuropsychological tests were transformed to z-scores, with higher scores corresponding to better performances.

## Lesions drawing

Patients' neuroimaging data was acquired via Computerised Tomography (CT, 85%) and Magnetic Resonance (MRI, 15%) and lesions were segmented and co-registered using the manual procedure already described by Moro and colleagues (*Moro et al., 2016*).

The lesion drawing was performed blindly and independently by two of the authors (VM, SB), prior (blind) to the group classification. In cases of disagreement on a lesion drawing, a third anatomist's opinion was consulted (<10%) and the different opinions were discussed until an agreement was reached.

Scans were registered on the ICBM152 template of the Montreal Neurological Institute, furnished with the MRIcron software (ch2, http://www.mccauslandcenter.sc.edu/mricro/mricron/). First the standard template was rotated on the three plans (size: 181 $\times$ 217$\times$181 mm, voxel resolution: 1 mm$^2$) in order to match the orientation of patient's MRI or CT scan. Lesions were outlined on the axial slices of the rotated template. The resulting lesion volumes were then rotated back into the canonical orientation, as to align the lesion volumes of each patient to the same stereotaxic space. Finally, in order to remove voxels of lesions outside the white and grey matter brain tissue, lesion volumes were filtered by means of custom masks based on the ICBM152 template.

## Disconnectome maps

Disconnectome maps were computed with the 'disconnectome map' tool of the BCBToolkit software (*Foulon et al., 2018*). The first step of the procedure is the tracking of white matter fibres passing through each patient's lesion, by means of the registration of lesions on the diffusion weighted imaging dataset of 10 healthy controls (*Thiebaut de Schotten et al., 2017*). This produces a percentage overlap map that takes into account the inter-individual variability of tractography in healthy controls' dataset (*Croxson et al., 2018*). Therefore, in the resulting disconnectome maps computed for each lesion, voxels show the probability of disconnection from 0% to 100% (*Thiebaut de*

*Schotten et al., 2015*). These disconnection probabilities of each patient are then used for statistical analyses.

## Statistical analysis producing the sites of lesion and tract disconnection

We ran two separate regression analyses for lesion sites and tract disconnections, using the same procedure. We used the tool 'randomize' (*Winkler et al., 2014*), part of FSL package (http://www.fmrib.ox.ac.uk/fsl/, version 5.0), which performs nonparametric statistics on neuroimaging data. Lesion drawings or disconnectome maps were considered as dependent variables within the general linear model implemented in 'randomize', in order to test the difference between the two groups in terms of disconnected brain regions. In these analyses white matter tracts and grey matter structures were not a-priori selected but the lesion and disconnection profiles of the whole right hemisphere were considered for each patient. Demographic (age, education), clinical (lesion size, lesion onset-assessment interval, motor deficit) and neuropsychological (personal and extra-personal neglect and memory impairment) data were considered in the model as control covariates. Threshold-Free Clusters Enhancement option was applied as to boost cluster-like structures of voxels and results that survived 5000 permutations testing were controlled for family-wise error rate ($p > 0.95$).

## Comparison of regression results with brain atlases

The involvement of the anatomical structures emerging as significant from the regression analysis results were compared with the probability maps of the insula, temporal pole and putamen (thresholded at 80%) of the Harvard-Oxford Atlas (*Makris et al., 2006*; *Desikan et al., 2006*). These masks were also used to compute the proportion of the insula, the Putamen and the TP affected by each patient's lesion.

In order to confirm the identity of the white matter tracts disconnected, we used an atlas of human brain connections (*Rojkova et al., 2016*). Results were compared with the probability maps (thresholded at 90%) of the cingulum, the frontal aslant (FAT) and the fronto-striatal tracts (FST) as well as the third branch of the superior longitudinal fasciculus (SLF III). Then, these masks were used to extract the probabilities of disconnection for each tract from each patient's disconnectome map. These probabilities were used to investigate the contribution of each tract and their disconnection co-occurrence.

## Bayesian post-hoc analyses

To test whether AHP emerges from the damage to grey matter structures and disconnection of each of these tracts independently or together as a whole, statistical analyses were conducted by using Bayesian models (R software, *R Development Core Team, 2018*; brms package, *Bürkner, 2017*) and generalised linear multilevel models were computed (Stan, *Carpenter et al., 2017*).

The presence of AHP (1) or its absence (0) was used as the dependent variable, while keeping demographic, clinical and neuropsychological variables as control covariates. As covariates of interest, we used proportion of the insula, the Putamen and the TP affected by each patient's lesion or the probability of each tract disconnection, ranged between 0 (=no lesion) to 1 (=full lesion).

As a first step, we tested whether the clinical/demographic model alone can predict the presence of AHP. Then, we explored whether the single grey regions or the single tracts can explain the presence of AHP better than the clinical/demographic model. We used the Bayes Factor ($BF_{10}$; 50): a $BF_{10}$ greater than three shows positive support for the hypothesis that the tract or grey matter structure is necessary, a $BF_{10}$ greater than 150 shows very strong support (*Raftery, 1995*).

After this, we fitted several binomial models, starting from the clinical/demographic model (i.e., with only the control covariates) to the full model (i.e. with all the clinical/demographic covariates, the grey regions, the tracts, and all the interactions among them), analysing the possible combinations of tracts and their interaction. The posterior samples were obtained by four chains, with 2500 burn-in and 2500 sampling iterations, resulting in a total of 10000 iterations for each posterior sample. Models including insula, temporal pole and putamen (i.e. all grey), fronto-striatal, frontal aslant tracts, cingulum and superior longitudinal fasciculus III (i.e. all white) as well as their full combination (i.e. all) were compared two by two.

## Data availability

The raw data used for this research (lesions) as well as the dependent variable and covariates are provided in full as *Source data 1*.

## Acknowledgements

We thank Lauren Sakuma for useful discussion and edits to the manuscript. This project has received funding from the European Research Council (ERC) under the European Union's Horizon 2020 research and innovation programme (grant agreement No. 818521 to MTS). This study was supported by a European Research Council Starting Investigator Award (ERC-2012-STG GA313755, to AF); Fondation pour la Recherche Médicale (FRM DEQ20150331725 to the frontlab team); by MIUR Italy (PRIN 20159CZFJK, to VM), and University of Verona (Bando di Ateneo per la Ricerca di Base 2015 project MOTOS, to VM).

## Additional information

### Funding

| Funder | Grant reference number | Author |
| --- | --- | --- |
| Horizon 2020 Framework Programme | 818521 | Michel Thiebaut de Schotten |
| Horizon 2020 Framework Programme | 313755 | Aikaterini Fotopoulou |
| Ministry of Education, Universities and Research | PRIN 20159CZFJK | Valentina Moro |
| University of Verona | Bando di Ateneo per la Ricerca di Base2015 project MOTOS | Valentina Moro |

The funders had no role in study design, data collection and interpretation, or the decision to submit the work for publication.

### Author contributions

Valentina Pacella, Conceptualization, Data curation, Formal analysis, Investigation, Methodology, Writing—original draft, Project administration, Writing—review and editing; Chris Foulon, Software, Formal analysis, Investigation, Methodology, Writing—review and editing; Paul M Jenkinson, Conceptualization, Data curation, Formal analysis, Writing—review and editing; Michele Scandola, Statistical analyses, writing-review and editing; Sara Bertagnoli, Renato Avesani, Resources, Investigation, Writing—review and editing; Aikaterini Fotopoulou, Conceptualization, Resources, Data curation, Investigation, Methodology, Writing—original draft, Writing—review and editing; Valentina Moro, Conceptualization, Resources, Data curation, Formal analysis, Supervision, Investigation, Methodology, Writing—original draft, Project administration, Writing—review and editing; Michel Thiebaut de Schotten, Conceptualization, Data curation, Software, Formal analysis, Supervision, Validation, Investigation, Visualization, Methodology, Writing—original draft, Project administration, Writing—review and editing

### Author ORCIDs

Chris Foulon (iD) https://orcid.org/0000-0002-7822-2653
Michel Thiebaut de Schotten (iD) https://orcid.org/0000-0002-0329-1814

### Ethics

Human subjects: All patients gave written, informed consent and the research was conducted in accordance with the guidelines of the Declaration of Helsinki (2013) and approved by the Local Ethical Committees of each center (see p. 10 l. 206-208).

Decision letter and Author response
Decision letter https://doi.org/10.7554/eLife.46075.008
Author response https://doi.org/10.7554/eLife.46075.009

## Additional files

### Supplementary files

• Source data 1.
DOI: https://doi.org/10.7554/eLife.46075.005

• Transparent reporting form
DOI: https://doi.org/10.7554/eLife.46075.006

### Data availability

The raw data used for this research (lesions) as well as the dependent variable and covariates are provided in full as Source data.

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
