## [Decision Letter]

Thank you for submitting your article "Anosognosia for Hemiplegia as a tripartite disconnection syndrome" for consideration by *eLife*. Your article has been reviewed by two peer reviewers, and the evaluation has been overseen by Laurel Buxbaum serving as Reviewing Editor and Richard Ivry as the Senior Editor. The reviewers have opted to remain anonymous.

The reviewers have discussed the reviews with one another and the Reviewing Editor has drafted this decision to help you prepare a revised submission.

Summary:

The investigators used structural images with probabilistic tractography to calculate disconnectome maps for 174 patients with right-hemisphere stroke. They identify three neural networks that contribute to anosognosia for hemiplegia (AHP) when disconnected: the (1) premotor loop (2) limbic system, and (3) ventral attention network. On the basis of these data, they suggest that motor awareness is dependent upon the joint contribution of these three networks.

The findings are of potential interest, represent a large sample, and may contribute to our understanding of the neural substrates of awareness of the status of the sensory-motor system. In that context, there are several concerns that need to be addressed.

Essential revisions:

A general point raised by the reviewers is that the study emphasizes white matter predictors of AHP at the expense of consideration of other potential predictors, and that a richer and more clinically useful approach would be to build the strongest possible predictive models given the available data. More specifically, the following concerns were noted:

1) What is the justification for why just 4 tracts were studied from among the plethora of white matter pathways in the brain? In particular, it would be useful to have information regarding the disconnection of arcuate, IFOF, SLF II, etc. A listing should be provided of the different tracts that were included in the connectome maps as well as the Bayes Factors scores for these.

2) The current manuscript overstates the significance of their findings by claiming that AHP is due to a disconnection syndrome rather than a combination of cortical and subcortical damage.

One way to resolve this is to combine both cortical and subcortical predictors in the same models. For example, a regression model can be performed in each cortical voxel that combines the lesion status of that voxel with its% connectivity (with every other brain voxel). The question is whether the combined predictive model results in a better fit to the AHP data than the disconnectome data. There are undoubtedly other approaches that would also enable assessment of the contribution of both cortical and subcortical regions to the pattern of behavior.

3) This issue in turn is related to the question of what (if anything) *fails* to predict AHP. Refer to Table 1. The patients with AHP were older, less educated, more acute, larger-lesioned, had more substantial neglect, a reduced digit span, and more hemiparesis. It is true that the investigators controlled for most of these factors in their statistical analysis (indicating that white matter tract disruption remains a strong predictor of AHP even in the context of these factors). However, they treated these important factors as "covariates of non-interest" and reported no results relevant to them. If indeed some of these factors contribute to the prediction of AHP, it tempers the enthusiasm we may have regarding the importance of knowing about tract integrity. Similar to the suggestion made with regard to point #1, more complex models should assess the contribution of these demographic and behavioral predictors to AHP, and then ask whether adding the connectome data further improves prediction of AHP.

---

## [Author Response]

Essential revisions:A general point raised by the reviewers is that the study emphasizes white matter predictors of AHP at the expense of consideration of other potential predictors, and that a richer and more clinically useful approach would be to build the strongest possible predictive models given the available data. More specifically, the following concerns were noted:1) What is the justification for why just 4 tracts were studied from among the plethora of white matter pathways in the brain? In particular, it would be useful to have information regarding the disconnection of arcuate, IFOF, SLF II, etc. A listing should be provided of the different tracts that were included in the connectome maps as well as the Bayes Factors scores for these.

We thank the reviewers and the editors for raising this point that allows us to better clarify the methods we used in the study.

“Why just 4 tracts were studied from among the plethora of white matter pathways in the brain?”

We did not select these four pathways in as a prior way but rather allowed the lesion data themselves to guide us. Specifically, our analysis strategy contains two steps. In the first step, we considered the whole right hemisphere disconnection profile of each patient (without any preselection of tracks) and computed a regression analysis to examine which disconnections predict anosognosia. This analysis revealed four tracks (i.e. Cingulum, SFL III, FAT, FST). In the second step, in order to establish the weight, i.e. contribution of each track to the syndrome, we entered only these same four tracts that were significant in the regression analysis of step one (i.e. Cingulum, SFL III, FAT, FST) as predictors in a Bayesian analysis able to determine the best ‘model’ fit for the syndrome. We are sorry we did not make this strategy clear in our original manuscript and we do so now in the Materials and methods section.

“It would be useful to have information regarding the disconnection of arcuate, IFOF, SLF II, etc.”

In order to follow the reviewers’ suggestion to control for the contribution of other tracts, as it may shed further light on the exclusive role of the four resulting tracts in AHP, we have now run some additional analyses on two tracts that were not significant in the regression analysis: i) the Inferior Fronto-Occipital Fasciculus and ii) the Superior Longitudinal Fasciculus II. Following the reviewers’ suggestion, we selected these two tracts as these are considered to play a role in symptoms that often co-occur with AHP, somatoparaphrenia (Gandola et al., 2012) and extra-personal neglect (Thiebaut de Schotten et al., 2012), respectively. We have followed the same procedure as above and we did not find these tracks to be significant at either step 1 or step 2 of our analysis strategy as we outline below.

In the Materials and methods section:

“We ran 2 separate regression analyses for lesion sites and tract disconnections, using the same procedure. […] In these analyses white matter tracts and grey matter structures were not a-priori selected but the lesion and disconnection profiles of the whole right hemisphere were considered for each patient.”

In addition, in the Results section we have now specified:

“No other tracts or structures were significantly involved in AHP”; “No other tracts contributed significantly to AHP.”

2) The current manuscript overstates the significance of their findings by claiming that AHP is due to a disconnection syndrome rather than a combination of cortical and subcortical damage.One way to resolve this is to combine both cortical and subcortical predictors in the same models. For example, a regression model can be performed in each cortical voxel that combines the lesion status of that voxel with its% connectivity (with every other brain voxel). The question is whether the combined predictive model results in a better fit to the AHP data than the disconnectome data. There are undoubtedly other approaches that would also enable assessment of the contribution of both cortical and subcortical regions to the pattern of behavior.

We are thankful to the reviewers for this point and in hindsight we agree that the emphasis we put on the role of white matter disconnections in the original manuscript may have overshadowed the contribution of grey matter structures, but this was mostly a matter of write-up as our analysis strategy (particularly Step 1) actually includes the investigation of both grey and white matter analyses and corresponding results (e.g. the results of grey matter lesion analysis revealed the significant contribute of several cortical and subcortical, grey matter structures such as the Insula, the temporal pole and the putamen to AHP). Hence, we have now revised the emphasis in our manuscript accordingly (please see below), as well as conducted further analyses at Step 2 in line with the comments made by the reviewers (please see below). Specifically, we investigated whether the addition of the above grey matter results (extracted from the analysis in Step 1), would also lead to a better model fit for the syndrome in our Bayesian analysis (Step 2).

Specifically, the mask of each of the three structures that resulted from the lesion analysis has been now produced from Insula, TP and Putamen probability maps of the Harvard-Oxford Atlas. Each map has been thresholded at 80% (i.e. the structure position in at least 80% of the healthy population), as a higher threshold did not return a sufficient number of voxels. These masks have been used to compute the proportion of the Insula, the Putamen and the TP affected by each patient’s lesion. By means of the Bayesian statistics, described on the response to point #1 and in detail in the previous version of the Results and Materials and methods sections, we have now compared the full model (i.e. the model including the clinical/demographic variables and the interaction among the disconnection probabilities of the Cingulum, the SLF III, the FAT and the FST) to a second model that included all the variables of the full model plus the proportion of the Insula, the TP and Putamen which was affected by the lesion (white+grey matter model). The results show a better fit of the second model compared to the first (BF_21_>150) supporting the hypothesis that the joint contribution of both disconnections and direct lesions within the three neural systems (premotor loop, limbic system, ventral attention network) better explains AHP. Overall this confirms that three neural networks contribute to AHP, when disconnected or directly damaged: the (1) premotor loop (2) limbic system, and (3) ventral attention network. We modified the text accordingly in the Abstract, Results, Figure 2 and legend, Discussion and Materials and methods section.

*3) This issue in turn is related to the question of what (if anything)* fails *to predict AHP. Refer to Table 1. The patients with AHP were older, less educated, more acute, larger-lesioned, had more substantial neglect, a reduced digit span, and more hemiparesis. It is true that the investigators controlled for most of these factors in their statistical analysis (indicating that white matter tract disruption remains a strong predictor of AHP even in the context of these factors). However, they treated these important factors as "covariates of non-interest" and reported no results relevant to them. If indeed some of these factors contribute to the prediction of AHP, it tempers the enthusiasm we may have regarding the importance of knowing about tract integrity. Similar to the suggestion made with regard to point #1, more complex models should assess the contribution of these demographic and behavioral predictors to AHP, and then ask whether adding the connectome data further improves prediction of AHP.*

In the existing literature, the debate about the specificity of AHP with respect to other cognitive functions is heated. AHP patients often present with a worse general cognitive profile than controls (Levine, 1990; Baier and Karnath, 2005; Cocchini et al., 2009), and a greater number of concomitant symptoms (Vocat, Staub, Stroppini, Vuilleumiere, 2010) and our sample is in line with these data. However, decades of work on anosognosia, as well as more recent large studies, have revealed single and double dissociations between anosognosia and each of these variables (e.g. Starkstein, Fedoroff, Price, Leiguarda, and Robinson, 1992, 1993; Bisiach, Vallar, Perani, Papagno, and Berti, 1986; Small and Ellis, 1996), as well as their combination (Marcel et al., 2004), supporting the possibility that these variables are not necessary for the presence of AHP, despite their frequent co-occurrence. For instance, relatively young patients, patients with small lesions and patients without neglect have been observed to show severe anosognosia in the above studies. Perhaps in the most studied concomitant symptom, namely neglect, the co-occurrence of neglect and anosognosia seems to be due to the presence of large lesions involving more than one network rather than to a functional association (Berti et al., 2006). This considered, and given the objective of our study, we decided to take into account these clinical and demographic variables in all the steps of our analyses: i) within the regression analysis by controlling for their effect; ii) in the Bayesian models starting from a model that only considers these clinical and demographical variables (erroneously called in the first version ‘Null model’). Indeed, this first model indicated a contribution of clinical and demographical variables and we apologize for not reporting this data in the first version. However, this is absolutely not in contrast with neuroanatomical data and confirmed by our results in Bayesian models, where the grey and white matter contribution are always controlled for demographic variables and where the neuroanatomical models result to be more predictive than the clinical-demographical one.

Specifically, the model with only the clinical/demographic variables is less predictive in explaining AHP: i) when compared to the model that includes each of the tracts resulting from the disconnection analysis (Cingulum, SLF III, FAT and FST), ii) when compared to the model that includes the interaction among the four tracts (full model) and iii) when compared to the models regarding grey matter structures and finally iv) when compared to the model that includes both white and grey matter structures. We can thus consider that, although often associated with other cognitive deficits, the specificity of the syndrome is associated with the damage of the three neural networks (premotor loop, limbic system, ventral attention network) described in the manuscript.

The text has been now changed in various points as follows:

Introduction: “Early studies regarded AHP as secondary to other concomitant symptoms (Cocchini et al., 2009; Vocat et al., 2010; Levine, 1990; Karnath, Baier and Nagele, 2005)”.

Results: “It is worth noting that although the starting model showed that the clinical and demographic variables alone contribute to the AHP symptoms, the model with only these control variables was less efficacious in predicting AHP: i) when compared to the models that included each of the tracts resulting from the disconnection analysis (Cingulum, SLF III, FAT and FST); ii) when compared to the model that included the interaction among the four tracts and iii) when compared to the model that included also the grey matter structures resulting from the lesion analysis (i.e. insula, Putamen and temporal pole, the white + grey matter model; BF>150). We can thus consider that, although the severity of concomitant cognitive deficits may contribute to AHP (Cocchini et al., 2009; Vocat et al., 2010; Levine, 1990; Karnath, Baier and Nagele, 2005), the syndrome is fully explained only when considering the integrated contribution of specific neural networks.”

Discussion: “Importantly, no isolated pattern of lesions or disconnections to any of these systems (i.e. the limbic system, the ventral attentional network, or the premotor system), or any demographical and clinical variables can fully explain AHP (Figure 2). […] After this, we fitted several binomial models, starting from the clinical/demographic model (i.e., with only the control covariates) to the full model (i.e. with all the clinical/demographic covariates, the grey regions, the tracts, and all the interactions among them),”